# A Review of Metamaterials in Wireless Power Transfer

**DOI:** 10.3390/ma16176008

**Published:** 2023-08-31

**Authors:** Cancan Rong, Lihui Yan, Long Li, Yunhui Li, Minghai Liu

**Affiliations:** 1Jiangsu Province Laboratory of Mining Electric and Automation, China University of Mining and Technology, Xuzhou 221008, China; ccrong@mail.cumt.edu.cn (C.R.); ts22230058a31@cumt.edu.cn (L.Y.); 2Key Laboratory of High-Speed Circuit Design and EMC of Ministry of Education, School of Electronic Engineering, Xidian University, Xi’an 710071, China; lilong@mail.xidian.edu.cn; 3School of Physical Science and Engineering, Tongji University, Shanghai 200092, China; liyunhui@tongji.edu.cn; 4State Key Laboratory of Advanced Electromagnetic Engineering and Technology, Huazhong University of Science and Technology, Wuhan 430074, China

**Keywords:** wireless power transfer, metamaterials, power transfer efficiency, misalignment tolerance, electromagnetic shielding

## Abstract

Wireless power transfer (WPT) is a technology that enables energy transmission without physical contact, utilizing magnetic and electric fields as soft media. While WPT has numerous applications, the increasing power transfer distance often results in a decrease in transmission efficiency, as well as the urgent need for addressing safety concerns. Metamaterials offer a promising way for improving efficiency and reducing the flux density in WPT systems. This paper provides an overview of the current status and technical challenges of metamaterial-based WPT systems. The basic principles of magnetic coupling resonant wireless power transfer (MCR-WPT) are presented, followed by a detailed description of the metamaterial design theory and its application in WPT. The paper then reviews the metamaterial-based wireless energy transmission system from three perspectives: transmission efficiency, misalignment tolerance, and electromagnetic shielding. Finally, the paper summarizes the development trends and technical challenges of metamaterial-based WPT systems.

## 1. Introduction

With the continuous development of science and technology, the traditional charging mode relying on the physical contact between conductors has gradually exposed many problems, especially in the fields of biomedicine, underwater power supply, transportation, and other fields. The traditional charging mode has disadvantages such as low safety, high maintenance cost, the easy aging of batteries, and so on. Wireless power transfer (WPT) technology is a kind of electric energy transmission technology that realizes no physical contact with soft media such as electric fields, magnetic fields, and electromagnetic waves. It has been rated as one of the top ten emerging technologies for two consecutive times. In 1891, Tesla first proposed the concept of WPT [1]. In the early 1960s, W. C. Brown made a lot of research on radio energy transmission and first demonstrated the microwave power transfer (MPT) system with electromagnetic radiation, which made this concept become a reality [2]. In 2007, Professor Marin Soljacic’s research group from the Massachusetts Institute of Technology proposed a magnetic coupling resonant wireless power transfer (MCR-WPT), which realized the wireless transmission of electric energy via a strong coupled magnetic resonance between two metal coils, becoming another milestone in the history of radio energy transmission [3]. Since then, more and more scholars have devoted themselves to the research of WPT technology. With the continuous development of science and technology and the needs of daily life, WPT technology has been widely used in various fields, such as electric vehicles [4,5,6,7,8], drones [9,10], portable electronic equipment [11,12], implant medicine [13,14,15,16], etc.

WPT technology can be divided into two types based on the distance from the electromagnetic field source: near-field coupling and far-field radiation [17]. As shown in Figure 1, near-field coupling includes magnetic induction coupling (MIC) WPT, magnetic coupling resonance (MCR) WPT, and electric coupling (EC) WPT. Far-field radiation includes microwave WPT, laser WPT, and ultrasonic WPT. MCR-WPT has advantages such as high transmission efficiency, long transmission distance, and high transmission power, making it the most practical and promising WPT technology compared to other near-field wireless power transfer technologies [18,19,20]. However, as the transmission distance increases, the coupling between coils decreases sharply, leading to a decrease in the system’s transmission efficiency. In recent years, many scholars have conducted research to address the limitations of the transmission distance and the transmission efficiency of wireless power transfer systems. Due to the loose coupling structure between the transmitting and receiving coils, there is an increase in reactive power in the system due to a significant leakage inductance, leading to a decreased transmission efficiency. Refs. [21,22,23,24] reduces the reactive power in the system by designing a compensation network to improve the transmission efficiency. Refs. [25,26,27,28,29] improves the coupling performance of the system by designing the magnetic coupling mechanism, increasing the magnetic field utilization, and hence, improving the transmission efficiency and offset tolerance. Additionally, the transmission distance and the efficiency of the system can be increased by using methods such as relay coils [30,31] and domino resonators. Each of the above methods has its advantages in improving the transmission efficiency, distance, and offset tolerance of the system, but there are also drawbacks such as the complex structure design and low practicality.

The loose coupling between WPT systems not only reduces the transmission efficiency of the system, but also leads to increased magnetic leakage around the system, and the surrounding electromagnetic radiation poses a serious threat to personal safety. For the electromagnetic compatibility of WPT systems, the International Commission on Non-Ionizing Radiation Protection (ICNIRP) has developed electromagnetic safety standards [32]. In recent years, electromagnetic compatibility has been a hot issue for WPT. The conventional way of electromagnetic shielding is to shield the system by adding a ferromagnetic core such as ferrite, but the ferrite is large and limits the design of the coupling mechanism [33,34]. There is also the electromagnetic shielding of the system by adding non-ferromagnetic metals, but the eddy current losses in non-ferromagnetic materials are high and the heating of the metal can lead to reduced system safety when the transmission power is high [35]. In summary, the conventional method of electromagnetic shielding using ferrite and nonferromagnetic metals, which can effectively reduce the leakage of the system, but there are problems such as the difficulty in taking into account both the shielding performance and transmission efficiency and low safety.

Metamaterials are artificially engineered composite materials with extraordinary physical properties that have been extensively studied and developed in various fields, such as electromagnetics, acoustics, and structural mechanics. These materials exhibit subwavelength structures, where the unit size is much smaller than the operating wavelength. Under the subwavelength structures, metamaterials can be considered homogeneous materials. The effective dielectric constant and effective magnetic permeability of metamaterials can be obtained using the effective medium theory. Metamaterials composed of subwavelength structural units possess extraordinary physical properties such as the negative refractive index [36], the inverse Doppler effect [37], and a perfect lens [38]. In 1967, Soviet physicist Veselago, based on Maxwell’s equations, discovered the propagation characteristics of electromagnetic waves in metamaterials and first proposed the concept of left-handed materials [39]. It was not until the 1990s that Pendry et al. designed the split ring resonator (SRR) to prove the feasibility of realizing negative permittivity and negative permeability [40,41,42]. In 2001, Smith achieved the left-handed material in the microwave band based on the SRR structure and observed the phenomenon of the negative refractive index [43]. With the continuous deepening of research on metamaterials, it has been found that by adjusting the structural parameters of the metamaterial unit, the magnetic permeability and dielectric constant can be arbitrarily controlled, thus producing various strange physical phenomena.

Transmission efficiency and electromagnetic shielding have always been hot issues that need to be addressed in WPT technology. In recent years, researchers have found that the extraordinary physical properties of metamaterials can improve the coupling between resonant coils and enhance the performance of WPT systems. B. Wang et al. first introduced negative magnetic metamaterials into WPT systems, successfully improving the transmission efficiency of the WPT system [44,45]. Due to the excellent electromagnetic modulation properties of metamaterials, more and more studies have been conducted to intervene metamaterials into WPT systems to improve the transmission efficiency as well as the misalignment stability of the system [46,47,48,49,50]. In addition, metamaterials with near-zero magnetic permeability have been designed in [51,52], which can effectively shield the magnetic leakage around the WPT system. Metamaterials have shown good performance in improving the transmission efficiency of WPT and reducing the electromagnetic leakage in the surrounding environment. However, the practical application of metamaterials in WPT systems is affected by the structure size, placement position, resonance frequency, and other factors of the metamaterials.

This article presents an overview of the research status and benefits of WPT technology based on metamaterials, along with its limitations and prospects. The subsequent sections of the article are organized as follows:Section 2 briefly introduces the basic principles of WPT technology.Section 3 describes the application principles of metamaterials in WPT from two perspectives, namely, electromagnetic modulation and the circuit theory.Section 4 introduces the current unit structures and design methods of metamaterials.Section 5 provides a comprehensive review of WPT technology based on metamaterials from three aspects: transmission efficiency, misalignment tolerance, and electromagnetic shielding.Section 6 discusses the limitations and prospects of metamaterials applied in WPT systems.Section 7 summarizes the entire article.

## 2. Analysis of MCR-WPT

The MCR-WPT technology is based on the magnetic resonance principle, which uses the magnetic field as a medium to achieve a wireless power transfer between the transmitting and receiving coils operating at the same resonant frequency [53]. Table 1 compares the performance of different WPT technologies, and the MCR-WPT technology has the following advantages compared to other WPT technologies: MCR-WPT improves the transfer distance to the mid-range (cm~m), which overcomes the short distance limitations of EC-WPT and MIC-WPT; it has a higher transmission efficiency and power than EC-WPT; compared to far-field radiation technology, MCR-WPT technology has a wider application range, lower implementation difficulty, and lower equipment costs; and it is not affected by non-ferromagnetic non-metallic obstacles. Therefore, MCR-WPT technology is the most practical and promising WPT technology.

Figure 2 presents the transmission framework of the MCR-WPT system, which comprises a magnetic coupler, compensation networks at the transmitting and receiving ends, a high-frequency inverter circuit at the transmitting end, a rectifying and filtering circuit at the receiving end, and a load [59]. At the transmitting end, the public frequency current is converted into a high-frequency current via the inverter. The compensation network then drives the transmitting coil to generate a high-frequency magnetic field. At the receiving end, the receiving coil operates at the same frequency as the transmitting coil and provides power to the load via the compensation network, rectifier, and regulator modules. The coupling between the receiving coil and the transmitting coil by the magnetic coupler is loosely coupled, which leads to an increase in stray magnetic fields in the space, causing a decrease in the system’s transmission efficiency and electromagnetic safety issues [60]. The high-frequency electromagnetic field around the entire coupling mechanism affects the surrounding equipment and biological entities. Therefore, transmission efficiency and electromagnetic safety are the two major issues that need to be addressed in WPT systems.

In the following, we will analyze a simple two-coil MCR-WPT system using the equivalent circuit theory. The equivalent circuit theory [61] is based on the theory of mutual inductance and Kirchhoff’s laws, which constructs a circuit model between the transmitting and receiving coils to seek equivalent relationships and solve the system. The equivalent circuit diagram of the simple two-coil WPT system is shown in Figure 3. Based on Kirchhoff’s laws, we can obtain the system’s voltage and the current equations, as shown below:(1)$$\left[\begin{array}{c} V_{s}\\ 0 \end{array}\right]=\left[\begin{array}{cc} R_{1}+R_{S}+jX_{1} & j\omega M_{12}\\ j\omega M_{21} & R_{2}+R_{L}+jX_{2} \end{array}\right]\left[\begin{array}{c} I_{1}\\ I_{2} \end{array}\right]$$
where *I*_1_, *I*_2_ denote the loop currents in the transmitting and receiving coils, respectively. *M*_12_, *M*_21_ denote the mutual inductance between the transmitting and receiving coils, respectively, and $M_{12}=M_{21}=M$. $X_{1}=\omega L_{1}-1/\omega C_{1},X_{2}=\omega L_{2}-1/\omega C_{2}$.

By calculating the loop currents on each resonant coil, we can obtain the load power and the active power of the transmitting coil as follows:(2)$$\{\begin{array}{c} P_{L}=\frac{{\left(\omega M\right)}^{2}V_{s}^{2}R_{L}}{{\left[\left(R_{1}+R_{s}\right)\left(R_{2}+R_{L}\right)-X_{1}X_{2}+\omega^{2}M^{2}\right]}^{2}+{\left[\left(R_{1}+R_{s}\right)X_{2}+\left(R_{2}+R_{L}\right)X_{1}\right]}^{2}}\\ P_{S}=\frac{\left\{\left(R_{1}+R_{2}\right)\left[{\left(R_{2}+R_{L}\right)}^{2}+X_{2}^{2}\right]+\omega^{2}M^{2}\left(R_{2}+R_{L}\right)\right\}V_{s}^{2}}{{\left[\left(R_{1}+R_{s}\right)\left(R_{2}+R_{L}\right)-X_{1}X_{2}+\omega^{2}M^{2}\right]}^{2}+{\left[\left(R_{1}+R_{s}\right)X_{2}+\left(R_{2}+R_{L}\right)X_{1}\right]}^{2}} \end{array}$$

Then, the power transmission efficiency (PTE) of the system can be expressed as:(3)$$PTE=\frac{P_{L}}{P_{S}}=\frac{\omega^{2}M^{2}R_{L}}{\left(R_{1}+R_{s}\right)\left[{\left(R_{2}+R_{L}\right)}^{2}+X_{2}^{2}\right]+\omega^{2}M^{2}\left(R_{2}+R_{L}\right)}$$

From Equation (3), we can see that the transmission efficiency of the system can be optimized by controlling the load impedance and mutual inductance *M*. Once the system parameters are determined, we can obtain the mutual inductance value at resonance, as shown below:(4)$$M=\frac{\pi \mu_{0}\sqrt{N_{1}N_{2}}{\left(r_{1}r_{2}\right)}^{2}}{2d^{3}}$$
where *r*_1_ and *r*_2_ are the radii of the transmitting and receiving coils, respectively; *N*_1_ and *N*_2_ are the number of turns of the transmitting and receiving coils, respectively; *d* is the transmission distance of the system.

From Equation (4), since the mutual inductance between the coils is approximately proportional to 1/*d*^3^, as the transmission distance increases, the mutual inductance between the coils rapidly decreases, and the PTE of the WPT system sharply decreases accordingly. The mutual inductance M between the resonant coils also affects the system’s frequency. When the distance between the resonant coils decreases, the mutual inductance increases, and the system enters an over-coupled state. The equivalent impedance of the receiving coil to the transmitting coil increases, and the impedance shows non-pure resistive characteristics, leading to frequency-splitting phenomena in the system [62].

Moreover, Ref. [63] analyzed the transmission efficiency of a four-coil MCR-WPT system using the circuit theory and the coupling mode theory. Ref. [64] presents a method for impedance matching in a four-coil WPT system. Although the two-coil system has a simpler structure, the four-coil system allows for impedance matching by adjusting the distance between the coils without changing the system’s structural parameters.

For low-frequency WPT systems, Equation (3) is often used to calculate PTE. However, for high-frequency magnetic resonance wireless power transfer systems, scattering parameters can be used to accurately characterize the transmission efficiency of the system. Scattering parameters simplify the design of the peripheral circuit of the system and the calculation of the load characteristics, reducing the experimental difficulty. Scattering parameters can be directly measured using a network analyzer, and the PTE of the system can be calculated using a two-port network model [65], as shown below:(5)$$PTE=\frac{P_{out}}{P_{in}}=\frac{{\left|S_{21}\right|}^{2}\left(1-{\left|\Gamma_{L}\right|}^{2}\right)}{\left(1-{\left|\Gamma_{in}\right|}^{2}\right){\left|1-S_{22}\Gamma_{L}\right|}^{2}}$$
where $\Gamma_{L}$ is the reflection coefficient of the load port, $\Gamma_{L}=\left(Z_{L}-Z_{0}\right)/\left(Z_{L}+Z_{0}\right)$, $\Gamma_{in}$ is reflection coefficient of the input port, and $\Gamma_{in}=S_{11}+\left(S_{11}S_{21}\Gamma_{L}\right)/\left(1-S_{22}\Gamma_{L}\right)$.

If the system impedance matches, then the PTE of the system can be expressed as:(6)$$PTE=\frac{{\left|S_{21}\right|}^{2}}{1-{\left|S_{11}\right|}^{2}}={\left|S_{21}\right|}^{2}$$

## 3. Fundamental of Metamaterials

As an artificial composite material, metamaterials possess unique electromagnetic properties compared to conventional materials. The permittivity *ε* and permeability *μ* are two parameters that describe the electromagnetic properties of the materials, which can indicate the material’s response to electric and magnetic fields. As shown in Figure 4, metamaterials can be classified into four categories based on the polarity of permeability and permittivity [66]: double positive (*μ* > 0, *ε* > 0), double negative (*μ* < 0, *ε* < 0), mu-negative (*μ* < 0, *ε* > 0), and epsilon-negative (*μ* > 0, *ε* < 0). The vast majority of the natural materials are in the first quadrant, where the wave vector of electromagnetic waves propagating in these materials is a real number and the waves can propagate within the medium. The second and third quadrants represent the magnetic-negative and electric-negative materials, respectively, where the electromagnetic waves cannot propagate but exist in the form of evanescent waves. The fourth quadrant represents the double-negative materials, where the electromagnetic waves can propagate. By combining the constitutive equation of the medium with the MAXWELL equation group, the wave vector *k*, electric field vector *E*, and magnetic field vector *H* of electromagnetic waves in the material can be obtained as shown below:(7)$$\{\begin{array}{l} k\times E=\omega \mu H\\ k\times H=-\omega \varepsilon E\\ k\cdot E=0\\ k\cdot H=0 \end{array}$$

It can be seen from Equation (6) that the Maxwell equation still holds, but *k*, *E*, and *H* of the electromagnetic wave in the double negative material (*ε* < 0, *μ* < 0) satisfy the left-handed spiral relationship, hence the double negative material is also called a left-handed material. Left-handed materials exhibit a negative refractive index and a swift wave amplification. It is found that the negative refraction and swift wave amplification of metamaterials can be applied to WPT systems, and the transmission efficiency can be effectively enhanced by using negative magnetic metamaterials, where the coupling efficiency of WPT systems was firstly improved by using negative magnetic metamaterials in [44]. In recent years, more and more researchers have applied metamaterials to WPT systems, which has become an important research direction for WPT. In the following, we will illustrate the principles of metamaterials applied to WPT systems from two aspects: electromagnetic regulation and the circuit theory.

### 3.1. Negative Refractive Index

When electromagnetic waves propagate from one medium to another, refraction and reflection occwjur. This process satisfies the continuous boundary conditions of the electric and magnetic fields of electromagnetic waves. According to Snell’s law, we can obtain the relationship between the angle of incidence and the refractive index, as shown below:(8)$$n_{1}\sin\theta_{1}=n_{2}\sin\theta_{2}$$
where $n_{1}$ and $n_{2}$ are the refractive indices of two different media and $n=\pm \sqrt{\varepsilon \mu }$, while *μ* and *ε* are the effective magnetic permeability and effective dielectric constant of the material, respectively. $\theta_{1}$ and $\theta_{2}$ are the angles of incidence and refraction, respectively.

At deep subwavelengths, i.e., when the spatial dimensions of the electromagnetic system are much smaller than the wavelength corresponding to its operating frequency, the electric and magnetic fields are decoupled, and only one of the magnetic permeability or dielectric constants needs to be negative to achieve the negative refractive property at this time. According to the boundary conditions, Equation (8) can be transformed into:(9)$$\frac{\tan\theta_{1}}{\tan\theta_{2}}=\frac{\mu_{1}}{\mu_{2}}$$

The familiar phenomena of reflection and refraction occur when an electromagnetic wave passes from air into a conventional material, as illustrated in the Figure 5. In this case, the incident and refracted waves are located on opposite sides of the interface. However, when an electromagnetic wave passes from a normal material to a mu-negative metamaterial (MNG), the direction of *k* and the Poynting vector S = *E* × *H* are opposite, and the phase compensation causes the incident and refracted waves to appear on the same side, exhibiting a negative refraction of the electromagnetic wave.

### 3.2. Evanescent Wave Amplification and Magnetic Shielding

Research has demonstrated that the incorporation of MNG metamaterials in WPT systems can improve the transmission efficiency by amplifying the evanescent waves. Additionally, near-zero permeability metamaterials (MNZ) can shield the magnetic field in WPT systems effectively. By analyzing the electromagnetic field, the amplification and shielding mechanisms of metamaterials on evanescent waves can be better understood [67]. As illustrated in Figure 6, the metamaterial plate is unbounded along the y and z directions, and the incident electromagnetic wave is a TE-mode polarized wave (S-polarized wave). The metamaterial plate interacts strongly with the polarized wave, resulting in three non-zero components: *H_x_*, *H_z_*, and *E_y_*. When the TE wave is incident on the metamaterial plate, the tangential electric field and magnetic field in the different regions can be expressed as follows based on Faraday’s electromagnetic induction law:(10)$$E_{y}=E_{0}e^{ik_{z}z}\{\begin{array}{ll} e^{ik_{x}x}+Re^{-ik_{x}x} & x<0\\ Ae^{i\kappa_{x}x}+Be^{-i\kappa_{x}x} & 0\leq x\leq L\\ Te^{ik_{x}\left(x-L\right)} & x>L \end{array}$$
(11)$$H_{z}=\frac{E_{0}e^{ik_{z}z}}{\omega \mu_{0}}\{\begin{array}{ll} k_{x}\left(e^{ik_{x}x}-Re^{-ik_{x}x}\right) & x\leq 0\\ \kappa_{x}\left(Ae^{i\kappa_{x}x}-Be^{-i\kappa_{x}x}\right) & 0\leq x\leq L\\ k_{x}Te^{ik_{x}\left(x-L\right)} & x\geq L \end{array}$$
(12)$$H_{x}=-\frac{E_{0}e^{ik_{z}z}}{\omega \mu_{0}}\{\begin{array}{ll} k_{z}\left(e^{ik_{x}x}-Re^{-ik_{x}x}\right) & x\leq 0\\ \sqrt{k_{0}^{2}\left(\mu_{m}/\mu_{0}\right)-k_{x}^{2}}\left(Ae^{i\kappa_{x}x}-Be^{-i\kappa_{x}x}\right) & 0\leq x\leq L\\ k_{z}Te^{ik_{x}\left(x-L\right)} & x\geq L \end{array}$$
where *L* represents the thickness of the metamaterial plate, and *R* and *T* represent the reflection and transmission coefficients of the electromagnetic wave at the interface between the air and the medium, respectively. *A* and *B* denote the amplitudes of the forward and reverse waves of the thermal wave that is incident on the metamaterial, respectively. Additionally, *k_z_* and *k_x_* represent the components of the wave number of the incident electromagnetic wave in free space in the z-direction and x-direction, respectively, and are satisfied $k_{x}^{2}+k_{z}^{2}=k_{0}^{2}$. Additionally, $\kappa_{x}=\sqrt{k_{0}^{2}\left(\mu_{m}/\mu_{0}\right)-k_{z}^{2}}$ represents the x-direction components of the wave number inside the metamaterial, and these components of the wave number satisfy the following relationship. *E*_0_, *k*_0_ represent the complex amplitude and wave number of the incident electromagnetic wave, respectively.

The transmission and reflection coefficients are derived from the boundary conditions *x* = 0 and *x* = *L* as follows:(13)$$\{\begin{array}{c} T\left(\kappa \right)=\frac{1}{\cos\left(\kappa_{x}L\right)-i\frac{\left[k_{0}^{2}\mu_{m}\left(\mu_{0}+\mu_{m}\right)-k_{z}^{2}\left(\mu_{0}^{2}-\mu_{m}^{2}\right)\right]}{2\mu_{0}\mu_{m}k_{x}\kappa_{x}}\sin\left(\kappa_{x}L\right)}\\ R\left(\kappa \right)=\frac{\frac{i}{2\mu_{0}\mu_{m}k_{x}\kappa_{x}}\left[k_{0}^{2}\mu_{m}\left(\mu_{0}-\mu_{m}\right)+k_{z}^{2}\left(\mu_{m}^{2}-\mu_{0}^{2}\right)\right]\sin\left(\kappa_{x}L\right)}{\cos\left(k_{tm}L\right)-i\frac{\left[k_{0}^{2}\mu_{m}\left(\mu_{0}+\mu_{m}\right)-k_{z}^{2}\left(\mu_{0}^{2}-\mu_{m}^{2}\right)\right]}{2\mu_{0}\mu_{m}k_{x}\kappa_{x}}\sin\left(\kappa_{x}L\right)} \end{array}$$

When the permeability of the metamaterial tends to −1:(14)$$\{\begin{array}{c} \underset{\mu_{m}\to -1}{\lim}T\left(\kappa \right)=1\\ \underset{\mu_{m}\to -1}{\lim}R\left(\kappa \right)=0 \end{array}$$

When the permeability of the metamaterial tends to 0:(15)$$\begin{array}{l} \underset{\mu_{m}\to 0}{\lim}T\left(\kappa \right)=\{\begin{array}{ll} \frac{1}{1-i\frac{k_{0}L}{2}}, & k_{z}=0\\ 0 & ,k_{z}\neq 0 \end{array}\\ \underset{\mu_{m}\to 0}{\lim}R\left(\kappa \right)=\{\begin{array}{ll} -\frac{k_{0}L}{2i+k_{0}L}, & k_{z}=0\\ -1, & k_{z}\neq 0 \end{array} \end{array}$$

From Equations (14) and (15), it can be observed that when the equivalent permeability of the metamaterial approaches −1, the evanescent wave is amplified, resulting in the maximum transmission of the magnetic field. Conversely, when the equivalent permeability of the metamaterial tends towards 0, most of the magnetic field is reflected. Figure 7a shows that placing MNG on the transmission channel of the wireless energy transmission system can focus the magnetic field and enhance the coupling between resonators, thereby improving the system’s transmission efficiency. Meanwhile, Figure 7b shows that placing the MNZ on the periphery of the wireless energy transmission system can improve the electromagnetic environment around the WPT system and alleviate the electromagnetic leakage issues.

### 3.3. Magnetic Dipole Coupling Theory

When the operating wavelength of a magnetically coupled resonant wireless energy transmission system is significantly larger than its transmission distance, the system is considered to be in a quasi-static state. In such cases, the resonant coil of the system can be approximated via a magnetic dipole model [68]. Figure 8 shows a model of an inductively coupled circuit between two coils and a metamaterial plate. The medium between the two magnetic dipoles affects both the mutual inductance between the magnetic dipoles and the self-inductance of the magnetic dipoles themselves. If the medium is composed of metamaterials, then the special electromagnetic intrinsic parameters of the metamaterials can be utilized to bring the two magnetic dipoles into complete coupling. In [69], the mutual inductance between the magnetic dipoles with and without loading the metamaterial was deduced as shown in the following equation:(16)$$M_{21}^{vac}=-\frac{\mu_{0}A_{1}A_{2}}{4\pi }\frac{1}{{\left(2D\right)}^{3}}\phi_{L}\left(0,3,\frac{d}{2D}\right)=-\frac{\mu_{0}A_{1}A_{2}}{4\pi d^{3}}$$
(17)$$M_{21}=-\frac{\mu_{v}\mu_{0}A_{1}A_{2}}{4\pi }\frac{h}{a{\left(2\alpha D\right)}^{3}}\phi_{L}\left(-\frac{b}{a},3,u\right)$$
where *A*_1_, *A*_2_ is the area of the magnetic dipole and the relative magnetic permeability of the metamaterial, respectively; $d=d_{1}+d_{2}+D$ is the distance between the magnetic dipoles; and $u=\left(\alpha D+d_{1}+d_{2}\right)$. α is the ratio of each anisotropy of the electromagnetic intrinsic parameters of the metamaterial, where α, *a*, *b*, *h* can be expressed as:(18)$$\{\begin{array}{l} \alpha =\sqrt{\mu_{x}/\mu_{z}}\\ a={\left(\left(\alpha /\mu_{x}\right)+\left(1/\mu_{v}\right)\right)}^{2}\\ b={\left(\left(\alpha /\mu_{x}\right)-\left(1/\mu_{v}\right)\right)}^{2}\\ h=\left(4\alpha /\mu_{x}\mu_{y}\right) \end{array}$$

Function $\Phi_{L}$ can be expressed as:(19)$$\Phi_{L}\left(z,s,\alpha \right)=\sum \frac{z^{n}}{{\left(n+\alpha \right)}^{s}}$$

The strength of the enhanced coupling between the two magnetic dipoles after loading the metamaterial plate can be obtained by calculating the ratio of $M_{21}^{vac}$ and $M_{21}$. Ref. [70] employs the theory of magnetic dipole coupling to compute the field surrounding the coil of the WPT system. The impact of the addition of the super-surface on the system’s efficiency is analyzed subsequently. The theoretical analysis highlights the enhancement in the mutual inductive coupling of the WPT system due to the super surface. However, this theory is only applicable to planar metamaterials, and the operating wavelength of the system should be much larger than the transmission distance of the apparatus.

### 3.4. Magnetic Inductive Wave Theory

Magnetically induced waves (MIW) propagate only in certain magnetic metamaterials formed by inductively coupled resonant circuits [71]. Metamaterials exhibit wave propagation properties in addition to the ability to amplify evanescent. In the case of deep subwavelengths, metamaterial cells can be equated to simple RLC resonant circuits. When a cell induces a current, it excites the surrounding metamaterial cells to produce induced currents, and the mutual coupling between the cells carries out the energy transfer. Magnetic induction wave devices are widely used in WPT systems as they can provide energy to multiple receivers with only one transmitter, increasing the spatial freedom [72,73,74,75].

The dispersion equation of the metamaterial array was derived through analyzing the MIW in systems of different dimensions [76,77]. A one-dimensional metamaterial array was taken as an example (Figure 9), where each metamaterial unit was treated as an RLC resonant circuit. To simplify the analysis, only the interaction between adjacent metamaterial units was considered. When the transmitter excites the first metamaterial unit, the current flowing through the nth unit of the metamaterial can be expressed as:(20)$$I_{n}=I_{0}e^{-j(\beta -j\alpha )na}$$
where $I_{0}$ is a constant, α is the attenuation factor, and β is the phase factor. By using the dispersion relationship between the metamaterial units, α and β can be derived as shown below:(21)$$\alpha =\frac{1}{k_{m}Q\sin\left(\beta a\right)}$$
(22)$$\beta =\frac{1}{a}\arccos\left(\frac{\omega_{0}^{2}w^{2}-1}{k}\right)$$

The aforementioned equation demonstrates that as *Q* and *k_m_* increase, the attenuation factor decreases. This model is better suited for explaining the formation of the stopband, which reduces the efficiency of the system.

### 3.5. Equivalent Circuit Theory

The analysis and design of metamaterials are generally based on the effective medium theory, which does not account for the non-uniform behavior of the electromagnetic response of the unit cell in the near-field and replaces the metamaterial, made up of periodically arranged units, with a homogeneous continuous medium described via effective parameters such as effective magnetic permeability. The effective medium theory uses periodicity to extract the electromagnetic properties of the metamaterial and to design and analyze it. However, in practical applications, the metamaterial plate is affected by the truncation effects. Moreover, when the metamaterial is incorporated into the WPT system, the interaction between the transmitting and receiving coils will affect the electromagnetic response of the metamaterial. Therefore, to analyze the electromagnetic response of the entire system, the metamaterial is equivalent to a simple RLC resonant circuit using an equivalent circuit model, which simplifies the theoretical analysis of WPT systems based on the metamaterials.

Figure 10 depicts the structural diagram of a two-coil WPT system with metamaterials. By introducing the metamaterial between the two coils, Kirchhoff’s voltage law (KVL) equation for the WPT system incorporating the metamaterial plate can be derived using the circuit theory, as shown below:(23)$$\{\begin{array}{l} \left(R_{s}+j\omega L_{t}+R_{t}+\frac{1}{j\omega C_{t}}\right)I_{1}+j\omega M_{ti}I_{i}+j\omega M_{tr}I_{2}=V_{S}\\ j\omega M_{ti}I_{1}+\left(R_{i}+j\omega L_{i}+\frac{1}{j\omega C_{i}}\right)I_{i}+j\omega M_{ij}I_{j}+j\omega M_{ri}I_{2}=0\\ \left(R_{r}+j\omega L_{r}+Z_{L}+\frac{1}{i\omega C_{r}}\right)I_{2}+j\omega M_{ri}I_{i}-j\omega M_{tr}I_{1}=0 \end{array}$$

## 4. Structure and Design of Metamaterials

In recent years, the utilization of metamaterials in WPT systems has gained significant attention. The metamaterial structure’s design is essential in determining the material’s electromagnetic properties and resonant frequency. In this section, we present a comprehensive overview of the commonly used metamaterial structures in WPT systems and introduce the prevalent design approaches for metamaterial units. To provide a better understanding, we discuss the advantages and disadvantages of each structure and analyze their performance in WPT systems.

### 4.1. Structure of the Metamaterial

The classification of metamaterial structures into 1D, 2D, and 3D based on their dimensions is illustrated in Figure 11, which have been extensively used in WPT systems. MNG metamaterials were first employed by Wang et al. to enhance the coupling coefficient between the resonant coils, and they designed 3D metamaterials that increased the WPT system efficiency by 30% via experiments [45]. Subsequent research has investigated the use of 3D metamaterials in WPT systems [78,79,80,81]. The effectiveness of 2D metamaterials with a magnetic resonant field enhancement (MR-FE) on the transmission efficiency of WPT systems were compared in [82]. The results show that one-dimensional planar compact metamaterials are more widely used in WPT systems and that 2D metamaterials may not be suitable for practical WPT applications due to the associated losses and complexity. Experimental verification of the effect of 1D, two-layer 1D, 2D, and 3D metamaterial plates on the efficiency of WPT systems was conducted in [83], with the efficiency improvement ranking of the four structures on the WPT system being 3D > two-layer 1D > 1D > 2D > original system. However, it should be noted that the excessive volume of metamaterial may not be practical for WPT applications due to the associated losses. Hence, one-dimensional planar compact metamaterials have gained wider acceptance in WPT systems.

The use of metamaterials in WPT systems must consider their volume, frequency, and loss. The most commonly used metamaterial structure is the metallic graphic structure based on Pendry’s proposed LC resonant cell array, with split ring resonators (SRRs) and spiral resonators (SRs) as the most typical artificial structural units. In 2001, Smith et al. presented the first microwave metamaterial, using a proposed circular open resonant ring that resonated at the GHz level [43]. However, WPT systems generally operate in the frequency range of 10 kHz to nearly 200 MHz [84], which the simple SRR structure cannot satisfy the low-frequency requirements of metamaterials. Furthermore, the development of metamaterials for WPT systems focuses on miniaturization, low loss, lightweight, and low-frequency characteristics. To achieve this, researchers have improved the metamaterial unit structures based on the in-depth studies of SRRs. The parts of unit cells of metamaterials for WPT are concluded in Figure 12.

In [70], the capacitively loaded split ring resonators (CLSRRs) was proposed based on the original SRRs and achieved a frequency reduction by using a large capacitor load. Refs. [85,86] employed square spiral resonators (SSRs) as the unit cell structure for their metamaterial. Compared to the SRRs, the SRs have a larger inductance and offer greater capacitance between the adjacent spirals, enabling a lower-resonant frequency. Additionally, the SRs result in reduced radiation losses due to its higher Q-factor. Based on the SR structure, a metamaterial unit with a size of only 1/158 of the working wavelength was obtained by adding a lumped capacitor [87]. Although the resonant frequency of the metamaterial can be adjusted by adding a lumped capacitor, further miniaturization of the metamaterial unit is necessary for application in WPT systems. Refs. [45,88,89] employed double-sided spiral structures to achieve more compact and miniaturized unit cells for the metamaterial. Refs. [90,91] improved the metamaterial unit cell’s equivalent capacitance and inductance by connecting the upper and lower metal spirals through vias on a double-sided spiral structure. Double-sided spiral structures are commonly used for miniaturization in metamaterial design, but the frequency reduction effect and losses gradually decrease with an increasing number of spiral turns. Fractal structures are another option with advantages such as structural compactness and self-similarity [92]. Ref. [93] analyzed the mechanisms by which the Koch and Hilbert fractal structures affect the frequency and permeability properties, further optimizing the SRs in WPT systems, but the design is complex. The aforementioned designs of the metamaterial unit cells mainly focus on the MHz frequency range, and there are significant challenges in applying the existing high-frequency metamaterial design theories to the kHz-frequency metamaterials. Currently, most kHz-frequency metamaterial structures use a double-sided spiral structure with vias and added capacitors [94,95]. Further improvement of the structure and optimization design methods are necessary for a low-frequency metamaterial design.

To meet the demands of the different frequencies for WPT systems, multi-frequency metamaterials have been studied [96]. A dual-layer metamaterial unit with distinct spiral structures on its upper and lower layers was used in [50]. This allowed the unit to operate at 13.56 MHz and 27.12 MHz frequencies. A dual-frequency metamaterial unit could also be obtained by using a dual-ring nested structure [97]. Additionally, Ref. [98] developed a low-loss circular spiral split ring resonator (CSSRR) by increasing the length of the spiral and miniaturizing the structure. This metamaterial can achieve a negative permeability at multiple microwave frequency bands.

Additionally, a Cubic High-Dielectric Resonator was proposed in [99], but it is a three-dimensional structure with large geometric dimensions and limited practicality. Ref. [100] used the planar structure of CSSRR to obtain a high dielectric constant, eliminating the limitations brought by the three-dimensional structure in the WPT system. Ref. [101] designed a metamaterial unit with a ferrite helical structure, which improves the efficiency of the low-frequency WPT systems while maintaining a compact and low-loss structure, with a structure size only 1/10,000 of the operating wavelength. Ref. [102] proposed compact metamaterial units based on ferrite cores, which enable the WPT systems to achieve a higher mutual inductance and received power.

**Figure 12 materials-16-06008-f012:**
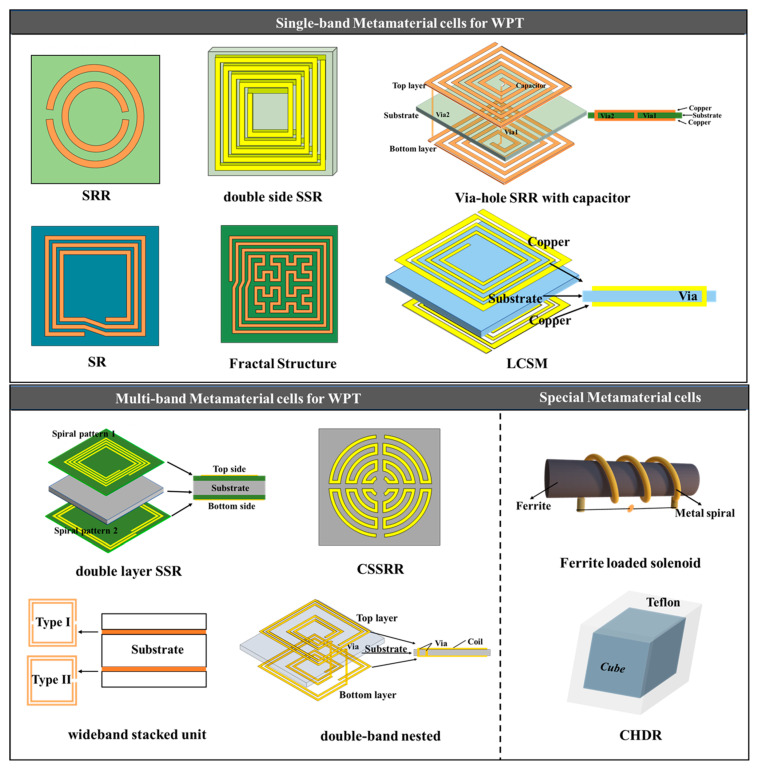
Unit cells of metamaterials for WPT [50,70,85,89,91,93,94,96,97,98,99,101].

Metamaterials derive their properties not from the properties of base materials, but from their specially designed structures. Their precise shape, geometry, size, orientation, and arrangement can affect light or sound or electromagnetic waves in an unusual manner, creating effects that are unachievable with conventional materials. The material composition of the metamaterials above is summarized in Table 2.

### 4.2. Design of the Metamaterial

Metamaterial design is primarily based on the effective medium theory, where the electromagnetic parameters and resonant frequency are the key considerations. These parameters can be manipulated by adjusting the structural parameters of the metamaterial unit cell. The S-parameter inversion method is commonly used in metamaterial design, where the desired electromagnetic properties can be obtained by adjusting the structural parameters. Additionally, some researchers have analyzed the equivalent circuit of metamaterials to obtain the parameters for simple and regular resonant units.

#### 4.2.1. S-parameter Inversion Method

In 2005, Smith et al. introduced the S-parameter inversion method as a means of calculating the intrinsic electromagnetic parameters of metamaterials based on the transmission coefficient *S*_21_ and the reflection coefficient *S*_11_ when plane waves are incident on the surface of the metamaterial [103]. This method is founded on the theory of equivalent medium, wherein the inhomogeneity of the electromagnetic response of the primitive in the space adjacent to the primitive can be disregarded when the size of the metamaterial primitive is considerably smaller than the electromagnetic wave wavelength. Consequently, the metamaterial comprised of periodically arranged cells can be substituted via a uniform continuous medium described using the equivalent magnetic permeability and other intrinsic parameters.

The S-parameter inversion process involves solving for the electromagnetic parameters of the metamaterial by constructing the transport matrix and scattering parameters using the equivalent model of the metamaterial. By inverting the scattering parameters of the primitive, the impedance and refractive index can be obtained, as shown in Equations (23) and (24) [104]:(24)$$n=\frac{1}{kd}\cos^{-1}\left[\frac{1}{2S_{21}}{\left(1-S_{11}^{2}+S_{21}^{2}\right)}^{2}+2\pi m\right]$$
(25)$$z=\pm \sqrt{\frac{{\left(1+S_{11}\right)}^{2}-S_{21}^{2}}{{\left(1-S_{11}\right)}^{2}-S_{21}^{2}}}$$

The scattering parameters of metamaterials can be obtained via electromagnetic software simulations based on two main types of cell simulations that utilize the electromagnetic wave theory. The first method involves adding a wave excitation between the simulated metamaterial cells, where Perfect Magnetic Conductors (PMC) are used on the upper and lower boundary surfaces and Perfect Electric Conductors (PEC) are used on the left and right boundary surfaces, as shown in Figure 13a. This method corresponds to a fixed incidence angle of electromagnetic waves and is suitable when the metamaterial has only finite period cells in the electromagnetic wave transmission direction. The second method involves applying Floquet ports to the upper and lower boundaries of the metamaterial and setting the master/slave boundary conditions on the front and rear boundary surfaces and the left and right boundary surfaces, respectively, as shown in Figure 13b. This method simulates a two-dimensional infinitely extended periodic arrangement structure and can adjust the direction of the electromagnetic wave incidence. Through these simulation methods, the scattering parameters of metamaterials can be obtained and used in the S-parameter inversion method to calculate the intrinsic electromagnetic parameters of the metamaterial.

The refractive index determination involves a complex logarithmic operation that may impact the real part of the refractive index results. To overcome this limitation, an enhanced algorithm based on the Kramers–Kronig (K–K) relation was proposed in [105]. This algorithm operates on a similar principle to the S-parameter inversion method and utilizes the intrinsic relationship between the imaginary and real parts of the analytic function to resolve the issue of branch selection and ensure the uniqueness of the sought value. This approach also improves the continuity of the refractive index across frequencies. However, it should be noted that the improved algorithm based on the K–K relation is only suitable for weakly coupled scenarios.

#### 4.2.2. Equivalent Model Method

The equivalent medium approach is a commonly used method for analyzing and designing metamaterials, but it has limitations when applied to systems operating in the near-field. Additionally, for low-frequency metamaterials, the constituent response is weak in the electromagnetic field simulations, which makes the parameter inversion methods unsuitable. In addition to analyzing and designing metamaterials from an electromagnetic field perspective, an equivalent model can be used to analyze and design metamaterials based on their resonance characteristics. The subwavelength structures can be equivalent to the simple resonant circuits, and the equivalent circuit can serve as a calculation model for the resonance characteristics of metamaterials in the optimization design. By utilizing this approach, we can achieve the optimization design of the metamaterial structure geometry parameters.

Ref. [106] introduces the concept of surface capacitance and equivalently models the circular metallic resonant ring as a resonant circuit composed of equivalent total capacitance and equivalent total inductance to obtain the resonant frequency. Ref. [107] focused on a novel strategy incorporating the dielectric effects into the thin-wire integro-differential formulation, enhancing the analysis of metamaterials. By considering the typically neglected substrate via a homogeneous equivalent medium approximation, it provides increased accuracy. Ref. [108] used accurate numerical simulations of the effect of resistive losses on the metasurfaces applied in the WPT system and analyzed the performance deviations brought about by the incorporation of the hypersurfaces in the system, providing ideas for the design of the metasurfaces. Ref. [59] analyzed the equivalent model of the metamaterial unit cells composed of square and circular spiral structures, which can better analyze the resonance characteristics such as the quality factor and the resonant frequency of metamaterials. Ref. [95] designs a metamaterial for the kHz band of the WPT system. The coupling between the systems was analyzed by equating the metamaterial cells to a simple resonant circuit. The metamaterial structure parameters were optimized via computational algorithms. The real part of the equivalent permeability curves at 85 kHz and 255 kHz is close to zero. Thus, the metamaterial cells can effectively reduce the leakage magnetic field at 85 kHz and 255 kHz.

## 5. WPT System Using Metamaterials

As previously mentioned, WPT technology has been widely applied in various fields but is limited by the transmission distance, efficiency, and electromagnetic leakage. Wang et al. approached this problem from a different angle and were the first to utilize the unique the physical properties of metamaterials to manipulate the magnetic field in WPT systems [44]. As discussed in the previous section, metamaterials can effectively control the magnetic fields, making them well-suited for WPT systems. Previous research has shown that metamaterials have three primary functions in WPT systems: improving efficiency, increasing fault tolerance, and shielding magnetic fields. In this section, we will provide a detailed introduction for the development of WPT systems based on metamaterials, with a focus on these three functions.

### 5.1. Efficiency and Distance Improvement

The efficiency of WPT systems is affected by the quality factor *Q* of the resonant coils and the coupling coefficient *k* between the coils, which determine the output power and transmission distance. Metamaterials, due to their unique electromagnetic properties, have the potential to improve the magnetic field distribution around the receiver, enhance the coupling between coils, and increase the transmission efficiency of the WPT system. To achieve an optimal performance, it is crucial to consider the structure, electromagnetic parameters, and placement position of the metamaterials used in WPT systems. These factors all play important roles in improving the efficiency of WPT systems based on metamaterials.

A metamaterial with near-zero permeability was designed in [15] to address the problem of performance degradation during the charging of wireless charging systems applied to biomedical implants. The metamaterial has minimal effect on the resonant frequency of the WPT system, and the efficiency of the system with the addition of the metamaterial is improved by more than 200% relative to that without the metamaterial when the transmission distance is 10 mm. Ref. [109] proposed a metamaterial-based WPT system for smart home applications. When the transmission distance is 50 cm, the PTE of the system with the addition of metamaterials is increased by 44.7%, and the operating frequency is 6.78 MHz. When the transmission distance is 140 cm, the PTE of the system with the addition of metamaterials is nearly 4.07 times higher than that of the original system, and the operating frequency is 433 MHz. The system is able to effectively increase the PTE of the system in both the near-field and the far-field. Ref. [90] designed a double-layer, thin PCB metamaterial with a single cell size of only 3.72 cm × 3.72 cm. The article also analyzed and optimized various parameters, including the number of turns of the metamaterial cell, the dielectric constant of the substrate, the thickness of the PCB board, and the unit cell battery substrate. Finally, a compact 5 × 5 array of metamaterial slabs was created based on the resonant coil’s size, successfully improving the WPT system efficiency from 17% to 47%.

In [110], a hybrid metamaterial slab (HMS) combining negative-magnetic metamaterials and zero-magnetic metamaterials was first proposed. The MNZ metamaterial units are used in the central unit of the metamaterial, while the MNG metamaterial units are used in the surrounding area. The MNZ metamaterial ensures that the magnetic field inside the WPT system propagates through the central line, and the MNG metamaterial concentrates the magnetic field passing through the edge of the HMS into the receiving coil. Additionally, a cubic structure with PEC and PMC boundaries was proposed in the paper to measure the real and imaginary parts of the magnetic permeability. The PTE increased from 34.5% to 41.7% when the transmitting and receiving coils were 15 cm apart. The leakage magnetic field decreased from −19.21 dBm to −26.03 dBm when the coils were 10 cm apart. In [111], the transmission efficiency of the WPT system was compared under different conditions, including no metamaterial slab, a slab of HMS, two slabs of HMS, and two slabs of MMs. The results showed that the efficiency of the system was significantly improved when two layers of HMS were added. Furthermore, for the design of the HMS slab, a trial-and-error method is commonly used to determine the physical and circuit parameters of the HMS slab. To overcome the limitations of this method, Ref. [112] proposed an experimental design-assisted sequential optimization method to change the structural parameters of the hybrid metamaterial slab to maximize its electromagnetic performance, thereby achieving a higher transmission efficiency of the WPT system. In [50], the WPT system integrating with the DB-MNG and DB-MNZ was proposed as shown in Figure 14. The implementation of DB-MNG in the system considerably enhanced the magnetic field around the receiver coil at 13.56 MHz and 27.12 MHz, respectively. As a result of the experiments, the efficiency of the system was increased by 15.44% and 7.69% at frequencies of 13.56 MHz and 27.12 MHz, respectively. Furthermore, the magnetic field density behind DB-MNZ was notably reduced.

Most of the research on WPT systems has concentrated on the placement of metamaterial plates in the middle of the resonant coil to enhance PTE. Although this approach can be effective, it reduces the flexibility and practicality of WPT systems. In [113], a compact double-layer metamaterial unit was designed to generate a smaller electric field than the traditional WPT systems, thereby enhancing the safety of the WPT system. Moreover, by placing the metamaterial plate close to the emitting coil, at a distance of 1 mm, the system’s efficiency was improved from 7% to 12% over a transmission distance of 25 mm. Placing the metamaterial slab in front of the resonant coil can integrate the resonant coils and the metamaterial slabs, but the strong coupling between them can lead to frequency shifts, splitting, and other phenomena which do not significantly enhance the transmission efficiency. In [114], two 4 × 4 metamaterial plates were placed on both sides of the resonant coil, and the simulation and experimental results demonstrated that the side-placed metamaterial plates significantly improved the transmission performance of the WPT system. When the transmission distance was 21 cm, the |S21| of the WPT system with the side-placed near-field metamaterial plate increased by 0.22. Placing the same metamaterial plate in the middle of the resonant coil did not enhance the transmission efficiency as much as placing it on the side. In [115], MNG and MNZ were side-placed in the WPT system using different combinations based on the principles of magnetic induction and magnetic shielding. The different combinations of the side-placed MNG and MNZ slabs are shown in Figure 15. When the transmission distance is 40 cm, and the distance D_1_ is 17 cm, the PTE is significantly improved, reaching 55.5%, 52%, and 51%, respectively, in Type B, C, and D. Side-placed metamaterials with various characteristics can increase the transmission efficiency of the WPT system and improve the leakage of the magnetic field around the system without affecting the transmission channel. However, side-placed metamaterial slabs are not appropriate for medium-to-long distance WPT systems.

### 5.2. Misalignment

A common issue in WPT technology research is the poor anti-offset capability. Most wireless power transmissions are currently based on single-coil-to-single-coil WPT, which necessitates the proper alignment of the two coils to achieve an efficient transmission. Position offset is a prevalent issue in practical WPT systems, and when the coupling mechanism is offset, both the efficiency and power will be greatly reduced. Several solutions have been proposed to address this problem, such as capacitance compensation and coil optimization [116,117].

In [100], a metamaterial plate with a high dielectric constant was designed and inserted in the WPT system. A schematic of the lateral misalignment and angular misalignment of the WPT system is shown in Figure 16a,b, respectively. The system’s efficiency is significantly improved by utilizing the strong magnetic dipole behavior produced by the high dielectric constant, even when the transmitting and receiving coils experience lateral or angular displacement, compared to the system without the metamaterial plate. In [48], a hybrid metamaterial plate with a tunable capacitor was used to achieve two different negative refractive index properties of −1 and −3. The capacitor value was adjusted based on the position of the receiving coil to allow the magnetic field to focus even when the transmitting and receiving coils were not aligned, improving the transmission efficiency by around 20% at a transmission distance of 70 cm. However, this tunable metamaterial has some limitations, such as only two capacitor values being designed to adjust the negative refractive index of the metamaterial unit, and the capacitor value needs to be changed autonomously, based on the position of the receiving coil. In [118], an anisotropic metamaterial was studied to address the issue of misalignment between the transmitting and receiving coils in a mid-range WPT system. The addition of the metamaterial plate increased the system’s efficiency by 30.9% when the receiving coil was angled at 45 degrees, effectively mitigating the decrease in PTE caused by misalignment.

### 5.3. Improving Electromagnetic Leakage

During the transmission process of WPT systems, the loose coupling between resonators can lead to electromagnetic leakage, which poses a security issue. Common methods for passive shielding include the use of ferrite and non-magnetic metals. While the addition of ferrite can effectively shield the leaked magnetic field, it is difficult to balance the transmission performance and is only suitable for low-to-medium frequency and low-power WPT systems [119,120]. The addition of non-magnetic metals results in high eddy current losses, which can affect the transmission efficiency of WPT systems. As discussed in Section 3, when the magnetic permeability of a metamaterial is zero or near-zero, it has the function of shielding the magnetic field. The use of metamaterials for magnetic field shielding can selectively shield specific frequency bands while simultaneously maintaining the system transmission efficiency.

In [121], the shielding principle of MNZ was analyzed in terms of electromagnetic waves using the Fresnel transmission and emission formulas. Placing a MNZ slab 10 cm behind the receiving coil, the experimental results showed that when the transmitting and receiving coils were 40 cm apart, the magnetic field intensity at the receiving coil was reduced by approximately 58.24%, which is a decrease of 13.14% compared with the original system. The combination of MNG with MNZ improved the efficiency of the WPT system by 12.06% and reduced the leakage magnetic field in the surrounding area. The original system’s magnetic field disperses in all directions upon passing through the receiving coil, resulting in significant magnetic field leakage. However, after placing an MNZ-EM board behind the receiving coil, the external magnetic field leakage is significantly reduced.

In [52], the working mechanism of MNZ in the WPT system was explained using the equivalent model theory. A human brain model was established using finite element simulation to verify the shielding effect of MNZ on the magnetic fields in the WPT system. The shielding effect of the metamaterial board was compared to that of a ferrite and an aluminum board via experiments. When the metamaterial is placed 10 cm behind the receiving or transmitting coil, and the human brain model is 10 cm away from the metamaterial board, the maximum magnetic field intensity in the human brain model is attenuated by 17.52 dB. The effects of the ferrite, the aluminum plate, and the MNZ slab placed at the transmitter terminal on the magnetic flux leakage of the system were compared. Compared with the ferrite and the aluminum plate, the electromagnetic noise of the transmitting terminal of the system with MNZ is reduced by 12.96 dB and 7.41 dB, respectively. The experimental results show that the MNZ slabs achieve the best shielding effect, resulting in a 9.5% improvement in the transmission efficiency of the system.

The previous studies on metamaterial-based WPT systems are summarized and compared in Table 3, where the normalized distance represents the ratio of the square root of the product of the transmission distance and the transceiver coil diameter.

## 6. Challenge and Prospect of Metamaterial in WPT Systems

In recent years, WPT technology has received widespread attention and application. Metamaterials, as materials with exotic physical properties, can promote the further development and application of WPT technology and potentially solve the bottleneck problems in current WPT technology. Although many studies have been conducted on the structure, placement, and frequency band of metamaterials applied in WPT systems, there are still many technical challenges and problems for WPT technology based on electromagnetic metamaterials:Design of miniaturized metamaterials for low-frequency applications. Currently, most low-frequency metamaterials operate in the MHz frequency range, which is too high for most electrical and electromagnetic devices. Moreover, the overall size of low-frequency metamaterials is large, which is not conducive to practical use in electromagnetic devices. The frequency range of electric vehicles and portable electronic devices is mostly in the kHz range. To enable metamaterials to meet the practical applications of WPT, it is necessary to study and design miniaturized metamaterials for low frequencies.Improve theoretical analysis. Existing simulation methods for microwave metamaterials are mostly based on the electromagnetic wave theory, but there is a lack of corresponding theoretical analysis for low-frequency metamaterials. Although low-frequency metamaterials have been designed using the equivalent circuit models, when the structure of metamaterials is complex, simple RLC resonant circuit models will be difficult to accurately describe their electromagnetic properties.Placement of metamaterials in WPT systems. Currently, most studies on metamaterial plates are placed between the receiving coil and the transmitting coil, which is very beneficial for improving the transmission efficiency of the system, but greatly reduces the practicality of metamaterials. The ideal placement position of metamaterials is fixed on the resonant coil, but when metamaterials are close to the transmitting coil, strong coupling between them will cause frequency splitting; when they are close to the receiving coil, the magnetic fields dissipate during the transmission process from the transmitting coil to the receiving coil, which cannot guarantee the transmission efficiency of the system. Therefore, it is necessary to study the placement position of metamaterial plates for practical applications.Design of multi-frequency and wideband metamaterials. Currently, most metamaterials adopt resonant metamaterials, which can only operate at fixed frequencies. In practical applications, compared to single-frequency metamaterials, dual-frequency or even multi-frequency metamaterials can improve the transmission efficiency and electromagnetic safety of two or more frequencies simultaneously. To further expand the application scenarios of metamaterials and achieve the simultaneous transmission of energy and signals, it is necessary to study wideband metamaterials with exotic electromagnetic properties.Achieve dynamic electromagnetic control. Passive metamaterials often come with certain energy losses, and their ability for electromagnetic control is relatively limited, which restricts their practical application in engineering. Research on intelligent control metamaterials can compensate for the energy losses of the elements and meet the different electromagnetic control requirements of WPT systems.

## 7. Conclusions

This article provides a review of WPT based on metamaterials. Firstly, the working mechanism of the WPT system is briefly introduced, followed by the description of electromagnetic characteristics, structures, and design methods of metamaterials. The applications of metamaterials in wireless power transfer are discussed, and the technological challenges and prospects of utilizing metamaterials in WPT systems are also addressed. The objective of this article is to summarize the research status and advantages of WPT technology based on electromagnetic metamaterials, as well as the future development, prospects, and directions.

## Figures and Tables

**Figure 1 materials-16-06008-f001:**
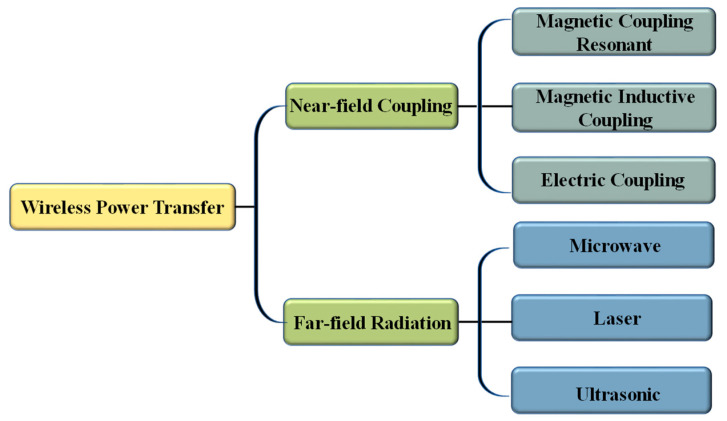
Classification of wireless power transfer.

**Figure 2 materials-16-06008-f002:**
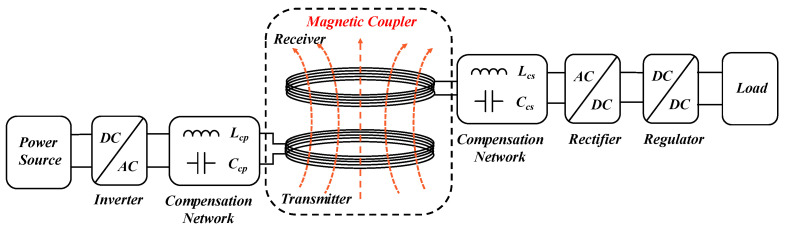
Composition of the MCR-WPT system.

**Figure 3 materials-16-06008-f003:**
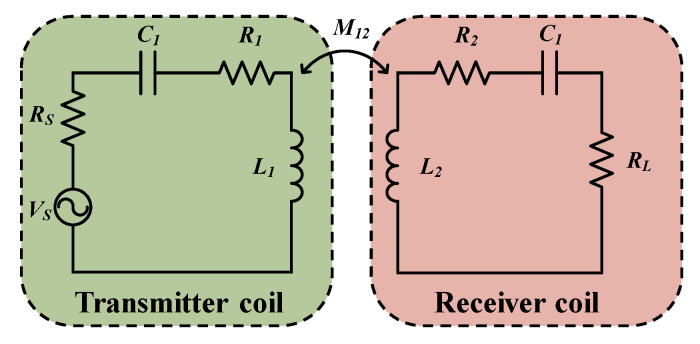
Simplified circuit model of a two-coil WPT system.

**Figure 4 materials-16-06008-f004:**
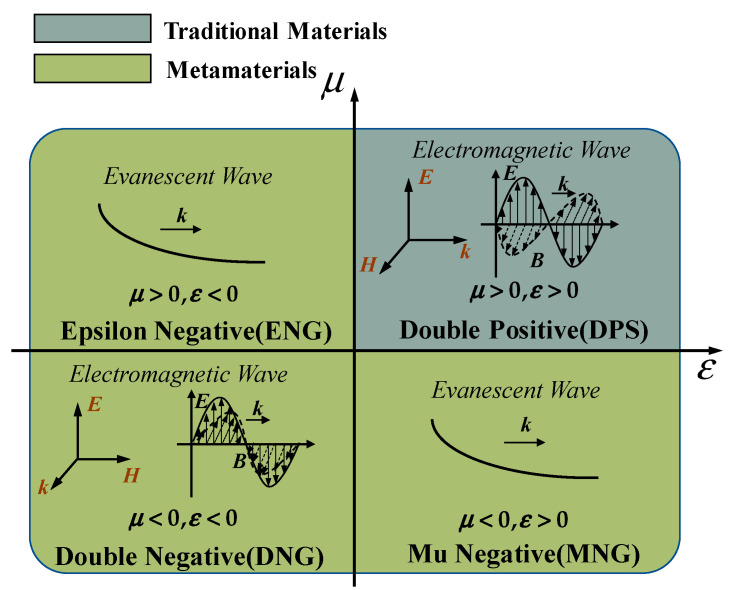
Classification of materials based on permeability and permittivity.

**Figure 5 materials-16-06008-f005:**
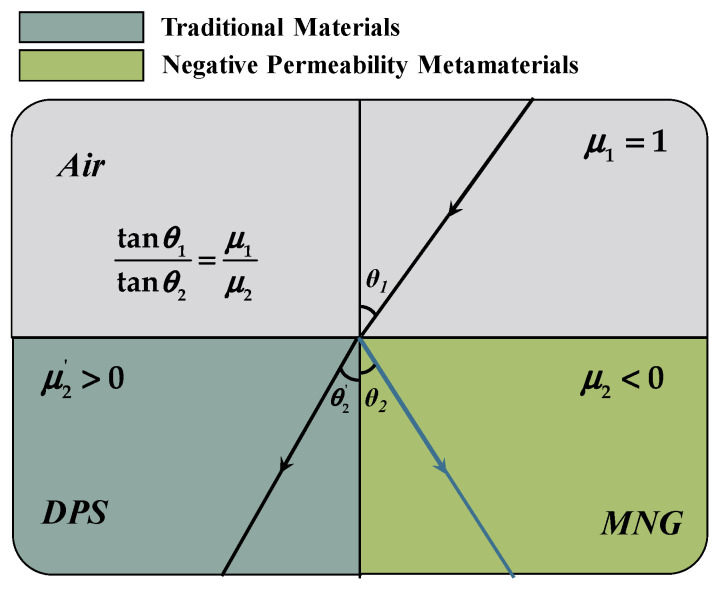
Refraction in the traditional material and metamaterial.

**Figure 6 materials-16-06008-f006:**
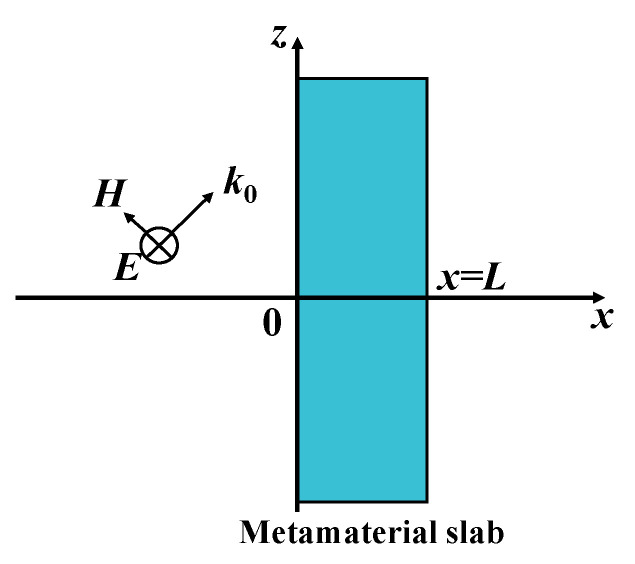
TE mode polarization wave incident on a metamaterial.

**Figure 7 materials-16-06008-f007:**
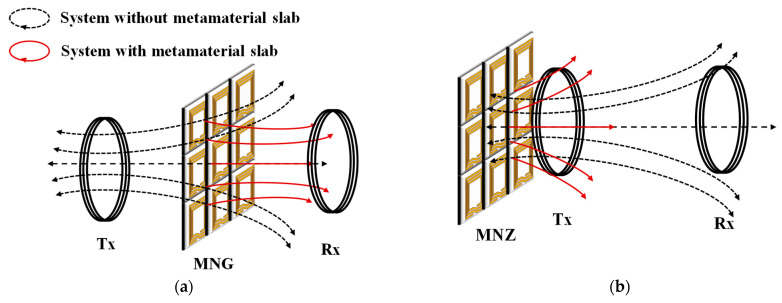
(**a**) Evanescent wave amplification with MNG; (**b**) magnetic shielding with MNZ.

**Figure 8 materials-16-06008-f008:**
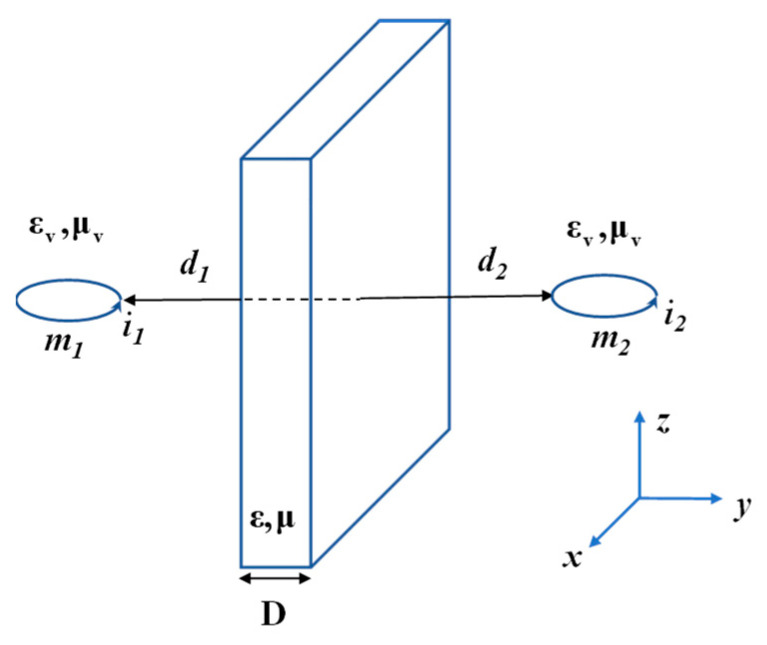
Magnetic dipoles separated by a metamaterial.

**Figure 9 materials-16-06008-f009:**
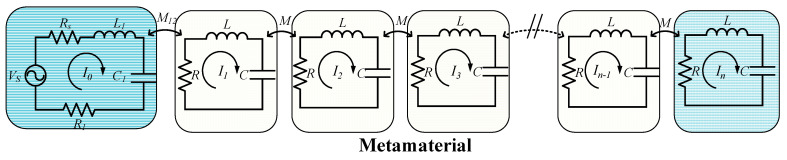
Equivalent circuit of 1-D metamaterial array in the WPT system.

**Figure 10 materials-16-06008-f010:**
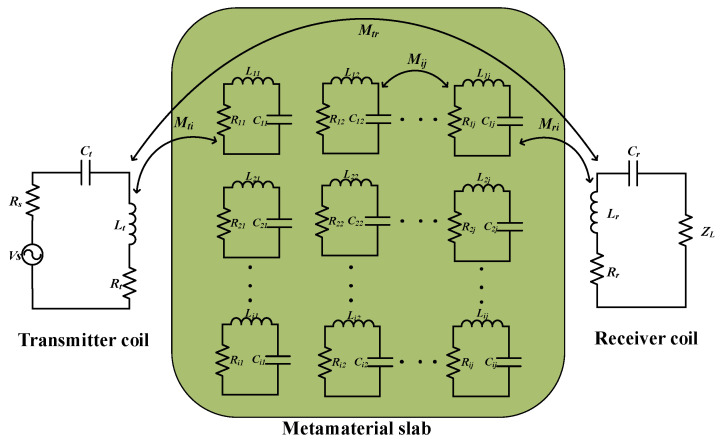
Equivalent circuit of a two-coil WPT system using metamaterials.

**Figure 11 materials-16-06008-f011:**
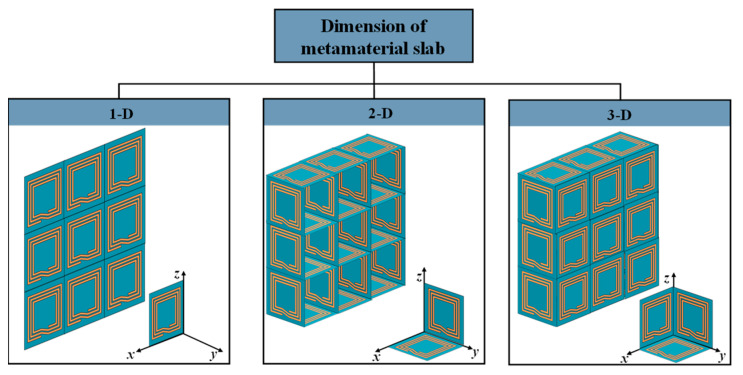
Classification of metamaterials by dimension.

**Figure 13 materials-16-06008-f013:**
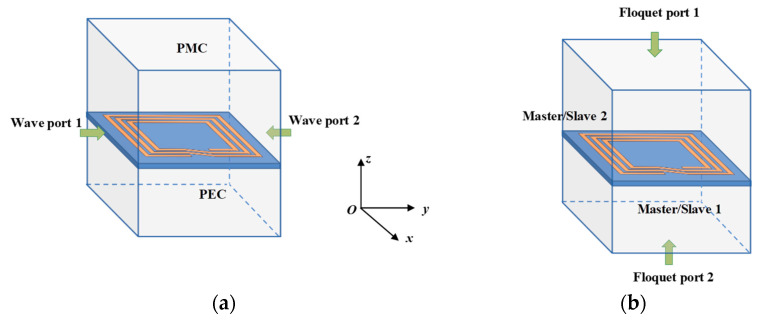
Electromagnetic software (ANSYS HFSS 2022 R1) simulations of metamaterial unit cells (**a**) Wave excitation ports; (**b**) Floquet ports.

**Figure 14 materials-16-06008-f014:**
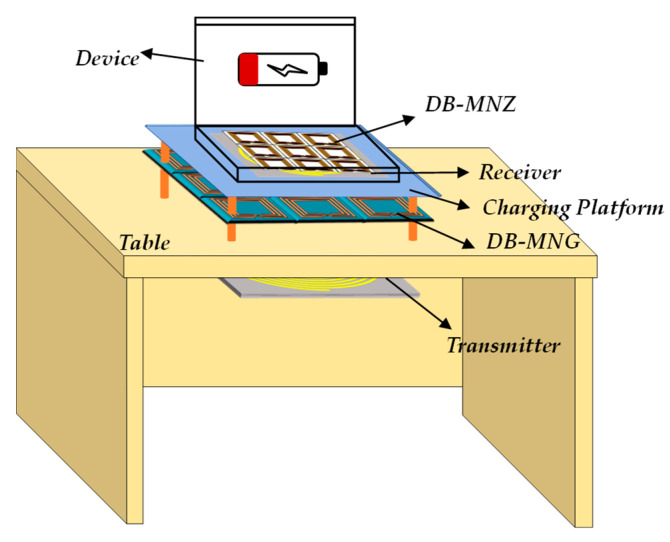
Schematic diagram of MRC-WPT system integrating DB-MNG and DB-MNZ for wireless charging of smart devices.

**Figure 15 materials-16-06008-f015:**
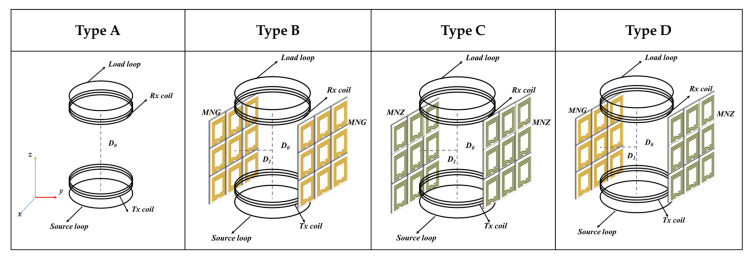
Illustration of the WPT system with different combinations of the side-placed MNG and MNZ slabs.

**Figure 16 materials-16-06008-f016:**
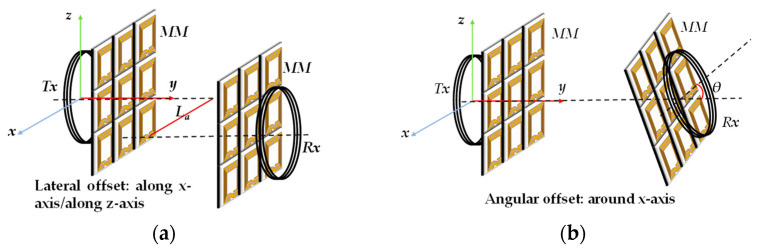
Illustration of different misalignment types of the WPT system using metamaterials: (**a**) Lateral misalignment; (**b**) angular misalignment.

**Table 1 materials-16-06008-t001:** Comparisons of different WPT technologies.

WPT Technologies		Power Range	Frequency	Transmission Distance	Transmission Efficiency
Near Field	MCR	100 W~1 MW	10 kHz~100 MHz	1 cm~1 m	70~95% [15,18]
	MIC	Tens of kW	Tens of kHz	1 mm~10 cm	80~95% [54]
	EC	1 W~3 kW	10 kHz~150 kHz	1 mm~10 cm	70~85% [55]
Far Field	Microwave	1 mW~3 MW	10 MHz~100 GHz	1 m~10 km	<40% [56]
	Laser	1 W~1 MW	>1 THz	1 m~10 km	<45% [57]
	Ultrasonic	0.1 mW~10 W	10 kHz~1 GHz	1 mm~5 m	<20% [58]

**Table 2 materials-16-06008-t002:** Composition of metamaterial and their properties.

Metamaterial	Base Materials	Properties	Reference
Split-Ring Resonators (SRRs) spiral resonators (SRs)	Metal (usually copper), substrate (often FR4)	Negative permeability behavior is attributed to the LC resonance. The preparation method based on PCB process has high precision, easy processing, good repeatability, and low cost.	[85,86]
Dielectric metamaterials	Ceramic samples, Teflon matrix	Negative permeability behavior is attributed to the Mie resonance. BST has high permittivity and low dielectric loss.	[99]
Ferrite based metamaterial	Ferrite, metal, or Teflon	Negative permeability behavior is attributed to ferromagnetic resonance of ferrite. Ferrite exhibits high quality factor, high inductance, and minimal losses.	[101,102]

**Table 3 materials-16-06008-t003:** Comparisons of different WPT systems using metamaterials based on previous research.

Position	Ref.	Structure of MMs	Operating Frequency (MHz)	Unit Cell Size Length × Wide × Thickness (mm)	Normalized Distance (cm)	PTE with/without MMs (%)	Shielding Effect
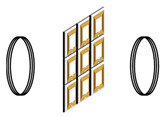 Middle	[45]	SR (single sided)	27.12	65 × 65 × 1.5	1.25	17/47	
	[90]	5 × 5 SR (double sided)	6.78	37.2 × 37.2 × 1.6	1.33	10.7/54.9	
	[110]	5 × 5 SR (single sided)	7.43	37.2 × 37.2 × 1.6	2.67	18.6/26.5	
	[48]	5 × 5 SR (single sided)	6.78	150 × 150 × 0.16	2	5.29/36.2 (displacement misalignment)	
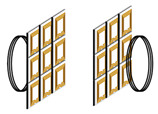 Front	[100]	4 × 4 CSSRR (single sided)	472.6	40 × 40 × 1.6	0.83	52.2/60.8	
	[113]	6 × 6 SR (double sided)	5.77	10 × 10 × 1	1.19	12/7	
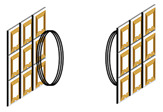 Back	[121]	3 × 3 SR (single sided)	13.56	120 × 120 × 1.6	2	36.24/48.3	58.24% (decrease)
	[52]	3 × 3 SR (single sided)	13.56	106 × 106 × 2	2.5	30.1/49.7	17.52 dB
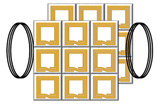 Side	[114]	4 × 4 LCSM (double sided)	6.78	47 × 47 × 1.6	1.57	7.4/40.3	
	[115]	4 × 4 SR (single sided)	13.56	120 × 120 × 1.6	2	55.5/52/51 (two NPM/two ZPM/one NPM and ZPM)	18.45 dBm Reduction (two ZPM)

## Data Availability

The data supporting the findings of this study are available by reasonable request to ccrong@mail.cumt.edu.cn.

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
