# Peer review of "A Review of Metamaterials in Wireless Power Transfer"

_materials, 2023, doi:10.3390/ma16176008_

Round 1

Reviewer 1 Report

The manuscript covers a review of metamaterials in wireless power transfer. Overall, it presents an overview of existing technology and explain the pros and cons clearly. Following are my comments and suggestions.

[1] In Table 1, please add some representative numbers of efficiency for each technology. (same for Table 2).

[2] What is the composition of metamaterial? Please include material information for some references. Authors may add a table of material properties of MM.

Reviewer 2 Report

This paper (Review paper) presents paper provides an 19 overview of the current status and technical challenges of metamaterial-based WPT systems. In my opinion this paper can be interesting to readers of Materials journal. The paper is not clearly presented. English of the paper is rather good and meet the requirement of the journal – in my opinion the language of the paper should be a little improved. The manuscript can be accepted for publication after MINOR corrections.

I find some editing mistakes for example:

-      Introduction chapter should be correct. Authors should include new information about topic of a paper. Amount of references is also sufficient but some papers cited in the references (80 from all 113) are older then 5 years. Author should include several modern papers of global research in this field – more information based on worldwide (global) study – not mostly from China (min 71 refereces). It would be desirable to expand this list somewhat by adding the work of other authors in the field of research over the past five years. That can help to emphasize the relevance and significance of this study.

-      Please refer to the scientific achievements of research teams from around the world (in particular from Europe and the USA) related to the topic of the paper.

-      Do the figures in the article belong to the authors - should there not sometimes be literature references next to them?

-      In figure captions for composite figures, the description should not start with a) (Figures 7, 15, 16 and 19). Each figure should have its own name, followed by (a), (b) etc.

The presented paper suits the requirements of the Materials journal and it may be publish after MINOR corrections.

English of the paper is rather good and meet the requirement of the journal – in my opinion the language of the paper should be a little improved.

Reviewer 3 Report

-The paper is a comprehensive review of the Metamaterial developments in the field of wireless power transference with valuable pedagogical features.

-As we deal with a review paper it is not so critical, but maybe authors could explain extensively about novelties and trends with respect to their 2021 work [73]:

Rong, C., et al.: A critical review of metamaterial in wireless power transfer systemIET Power Electron. 141541 1559 (2021).  https://doi.org/10.1049/pel2.12099

Some details:

-Figure 2 caption is a repetition of figure 1 which does not match the image

-Line 296. z=0, z=L   --> should not be  x=0 x=L ?

-Following eq. 10 and 11 one could expect expression for Hx

It is fine.
